# MV_Hybrid: Improving Spatial Transcriptomics Prediction with Hybrid State Space-Vision Transformer Backbone in Pathology Vision Foundation Models

**Won June Cho**[1], **Hongjun Yoon**[1], **Daeky Jeong**[1], **Hyeongyeol Lim**[1],
**Yosep Chong**[2]    WJCHO,HYOON,DKJEONG,HYLIM@DEEPNOID.COM, YCHONG@CATHOLIC.AC.KR

[1] *AI Research Team 2, AI Research Lab, Deepnoid*
*Seoul, Republic of Korea*
[2] *Department of Hospital Pathology, College of Medicine, The Catholic University of Korea*
*Seoul, Republic of Korea*

## Abstract

Spatial transcriptomics reveals gene expression patterns within tissue context, enabling precision oncology applications such as treatment response prediction, but its high cost and technical complexity limit clinical adoption. Predicting spatial gene expression (biomarkers) from routine histopathology images offers a practical alternative, yet current vision foundation models (VFMs) in pathology based on Vision Transformer (ViT) backbones perform below clinical standards. Given that VFMs are already trained on millions of diverse whole slide images, we hypothesize that architectural innovations beyond ViTs may better capture the low-frequency, subtle morphological patterns correlating with molecular phenotypes. By demonstrating that state space models initialized with negative real eigenvalues exhibit strong low-frequency bias, we introduce MV_Hybrid, a hybrid backbone architecture combining state space models (SSMs) with ViT. We compare five other different backbone architectures for pathology VFMs, all pretrained on identical colorectal cancer datasets using the DINOv2 self-supervised learning method. We evaluate all pretrained models using both random split and leave-one-study-out (LOSO) settings of the same biomarker dataset. In LOSO evaluation, MV_Hybrid achieves 57% higher correlation than the best-performing ViT and shows 43% smaller performance degradation compared to random split in gene expression prediction, demonstrating superior performance and robustness, respectively. Furthermore, MV_Hybrid shows equal or better downstream performance in classification, patch retrieval, and survival prediction tasks compared to that of ViT, showing its promise as a next-generation pathology VFM backbone. Our code is publicly available at: https://github.com/deepnoid-ai/MVHybrid.

**Keywords:** Vision Foundation Models, State Space Models, Computational Pathology

## 1 Introduction

Spatial transcriptomics (ST) technologies (Ståhl et al., 2016) have emerged as a powerful tool for understanding tissue biology by preserving both single-cell transcriptome and spatial context, which addresses key limitations of bulk RNA sequencing and single-cell RNA sequencing. This spatial resolution is particularly valuable in precision oncology research, where ST data can reveal complex patterns of tumors that can further improve patient

outcomes in the clinic—for example, through treatment response prediction and tumor microenvironment analysis (Elhanani et al., 2023; Hwang et al., 2022; Arora et al., 2023). However, the clinical adoption of ST remains limited by high costs, technical complexity, and the need for specialized tissue processing that disrupts standard pathology workflows (Zhang et al., 2022; Jin et al., 2024; Pentimalli et al., 2025). These barriers have motivated the development of deep learning approaches to predict spatial gene expression patterns directly (Xie et al., 2023; Zeng et al., 2021; He et al., 2020) from routine hematoxylin and eosin (H&E) stained whole slide images (WSI), which are already integral and commonly used in clinical diagnosis.

With the recent release of large-scale public ST-H&E WSI paired datasets (Jaume et al., 2024; Chen et al., 2024a) and the introduction of vision foundation models (VFMs) in histopathology (Chen et al., 2024b; Zimmermann et al., 2024; Xu et al., 2024; Saillard et al., 2024), biomarker prediction models (Zhu et al., 2025; Wang et al., 2024; Chung et al., 2024) use these pretrained VFMs in their training methods as they have captured diverse morphological features that correlate well with underlying molecular phenotypes. Indeed, through large-scale pretraining of Vision Transformers (ViT) (Dosovitskiy et al., 2021) using the DINOv2 (Oquab et al., 2023) self-supervised learning (SSL) method, these *state-of-the-art* VFMs have saturated multiple validation benchmarks in cancer subtype classification and detection (Campanella et al., 2025; kaiko.ai et al., 2024; Zhang et al., 2025) by showing clinical-level performance. However, Jaume et al. (2024) introduced HEST-Benchmark, a paired ST-H&E data across multiple cancer subtypes, which showed that current VFMs perform below clinical-grade in biomarker prediction via gene expression regression from patch embeddings. Campanella et al. (2025) and Zhang et al. (2025) also show similar results, meaning that biomarker prediction now serves as both a practical application and a rigorous benchmark for evaluating the representation power of these VFMs.

Furthermore, de Jong et al. (2025) and Kömen et al. (2024) show that pathology VFMs are unrobust and vulnerable to batch effects as they favor learning site (hospital)-specific features over true biological features. Given that these VFMs are pretrained on millions of diverse WSIs, we propose that the unrobustness may not be entirely due to data diversity itself, but partly due to the ViT backbone architecture of the VFMs. Likewise, Mao et al. (2025) revealed that WSIs, compared to natural images, contain much larger portions of high frequency features. While VFM downstream tasks like tumor detection and classification are mostly based on identifying human detectable high frequency features like tumor boundaries, biomarker prediction (ex. predicting expression of HER2) is inherently more difficult as it requires models to capture low-frequency features that are beyond human perception—the complex relationship between tissue morphology and underlying molecular states must be captured. Therefore, building on the work of Yu et al. (2025), which shows that state space models (SSMs) exhibit strong low-frequency bias, we design an SSM with an even stronger low-frequency bias and integrate it with ViT layers to form a hybrid state space-ViT model backbone, named $MV_{Hybrid}$, to replace ViT as the backbone in VFMs.

$MV_{Hybrid}$ consists of a SSM called MambaVision (MV) (Hatamizadeh and Kautz, 2025) in the first half of its layers and a ViT in the second half to learn more useful low-frequency biological features for biomarker prediction. We used DINOv2 to pretrain $MV_{Hybrid}$ and five other SSM and ViT models on publicly available colorectal cancer (CRC) datasets. While Nasiri-Sarvi et al. (2024) already showed the potential of SSMs by self-supervised

pretraining of Vision Mamba (ViM) (Zhu et al., 2024) to outperform ViT, it was only evaluated on a simple downstream classification task based on a single dataset and also did not consider other SSM backbone architectures. Therefore, our contributions are: 1. MV$_{\mathrm{Hybrid}}$ shows superior biomarker prediction and robustness ability compared to other models like ViT when evaluated on validation splits stratified by study sources. 2. MV$_{\mathrm{Hybrid}}$ also shows better performance in other tasks like classification, patch retrieval, and survival prediction, further showing its potential to be a strong candidate for future pathology VFM pretraining. 3. To this date, this work serves as the first paper in pathology VFMs where numerous VFM backbones are both pretrained and evaluated on the identical dataset.

## 1.1 Preliminary: State Space Models

Structured State Space Models (SSMs) represent a sequence-to-sequence transformation through a linear time-invariant (LTI) dynamical system. The continuous-time formulation is given by:

$$\frac{dx(t)}{dt} = Ax(t) + Bu(t) \tag{1}$$

$$y(t) = Cx(t) + Du(t) \tag{2}$$

where $A \in \mathbb{C}^{N \times N}$, $B \in \mathbb{C}^{N \times 1}$, $C \in \mathbb{C}^{1 \times N}$, and $D \in \mathbb{C}$ are learnable parameters, with $N$ being the state dimension. The state matrix $A$ governs the system's dynamics and frequency characteristics through the state evolution shown in equation (1)—its eigenvalues determine which frequencies are preserved or attenuated in the state evolution. To analyze the frequency response of SSMs, we derive the transfer function $G(is)$ (with $s$ representing frequency) by applying the Laplace transform to equations (1) and (2) with $s = i\omega$ for frequency analysis. Solving for the state $X(s) = (sI - A)^{-1}BU(s)$ from the transformed state equation and substituting into the output equation yields:

$$G(is) = C(isI - A)^{-1}B + D = \sum_{j=1}^{N} \frac{c_j}{is - a_j} + D \tag{3}$$

where $a_j$ are the eigenvalues of $A$ and $c_j = (CB)_j$ are the residues from partial fraction decomposition. Each term $\frac{c_j}{is - a_j}$ acts as a first-order low-pass filter whose cutoff frequency and behavior depend critically on the eigenvalue $a_j$.

**Frequency Bias in SSMs.** Yu et al. (2025) established that SSMs exhibit an inherent frequency bias, where the transfer function $G(is)$ has greater total variation in low-frequency regions than high-frequency regions. For a diagonal matrix $A = \mathrm{diag}(a_1, \ldots, a_N)$ with eigenvalues $a_j = v_j + iw_j$ where $v_j < 0$ (for stability), the frequency bias is quantified by the total variation $V_a^b(G)$, which measures how much the transfer function changes over the frequency interval $[a, b]$:

$$V_a^b(G) = \int_a^b \left| \frac{dG(is)}{ds} \right| ds \tag{4}$$

**SSM Variants.** Mamba (Gu and Dao, 2023) introduces selective parameters that become input-dependent, enabling dynamic state adjustment. ViM modifies Mamba to process sequences bidirectionally, and SiMBA (Patro and Agneeswaran, 2024) adds EinFFT channel

mixing layers to the Mamba sequence mixing layers for additional numerical stability during training. MV replaces the causal convolution layers in Mamba with regular convolutional layers and adds additional regular convolutional layers in the skip connection layer of the SSM block for enhanced visual processing capabilities. Hydra (Hwang et al., 2024) also replaces causal convolutional layers with regular convolutional layers but uses quasiseparable matrices (instead of Mamba's semiseparable) for natural bidirectional modeling.

**Enhanced Low-Frequency Bias of Negative Real Eigenvalues.** The Mamba variants MambaVision, ViM, and Hydra all use *negative real eigenvalues* $a_j = -|\lambda_j|$ where $\lambda_j > 0$. This choice provides *enhanced low-frequency bias* compared to complex eigenvalues. To understand why, recall that lower total variation at high frequencies means the system preserves low frequencies while suppressing high frequencies more effectively.

For complex eigenvalues $a_j = v_j + iw_j$, Yu et al. (2025) shows that the high-frequency total variation is bounded by:

$$V_{\omega_0}^{\infty}(G) \leq \sum_{j=1}^{N} \frac{|c_j|}{|w_j - \omega_0|} \tag{5}$$

where $\omega_0$ represents a high-frequency threshold. This bound arises from evaluating the integral of $|\frac{dG(is)}{ds}|$ from $\omega_0$ to $\infty$ (detailed derivation is in Appendices A.1 and A.2).

For negative real eigenvalues $a_j = -|\lambda_j|$, the high-frequency behavior can be approximated as (detailed derivation is in Appendix A.3):

$$V_{\omega_0}^{\infty}(G) \sim \sum_{j=1}^{N} \frac{|c_j|}{\sqrt{|\lambda_j|^2 + \omega_0^2}} \tag{6}$$

Here, $\omega_0$ represents a high-frequency threshold. For large $\omega_0$, this approximation shows that $V_{\omega_0}^{\infty}(G) \sim O(1/\omega_0)$, which decays faster than the complex eigenvalue case where $V_{\omega_0}^{\infty}(G) \sim O(1/(\omega_0 - w_j))$. This faster decay indicates less variation ($\frac{1}{\omega_0} < \frac{1}{\omega_0 - w_j}$) at high frequencies and thus *even stronger low-frequency bias*. The key advantage is that negative real eigenvalues create a *uniform frequency cutoff* around $|s| \approx \max(|\lambda_j|)$ (detailed description and analysis are in Appendix A.4), whereas complex eigenvalues have cutoffs distributed across different frequencies $|w_j|$.

## 2 Methods

We first detail the architecture of $\text{MV}_{\text{Hybrid}}$ and the architecture of other models followed by a description of experiments and datasets used.

### 2.1 Architecture of Pretrained Models

Figure 1 shows the architecture of our derived model, $\text{MV}_{\text{Hybrid}}$. This is a hybrid model because the first half (12 layers) of the model consists of a MV block (sequence mixing layer) and an EinFFT block (channel mixing layer) and the second half (12 layers) contain ViT layers. Since the original MV implementation consists of hierarchical backbones, we modify the backbone to be isotropic to make it suitable for DINOv2 pretraining. The architecture of other pretrained models is detailed in Table 1, where each model contains

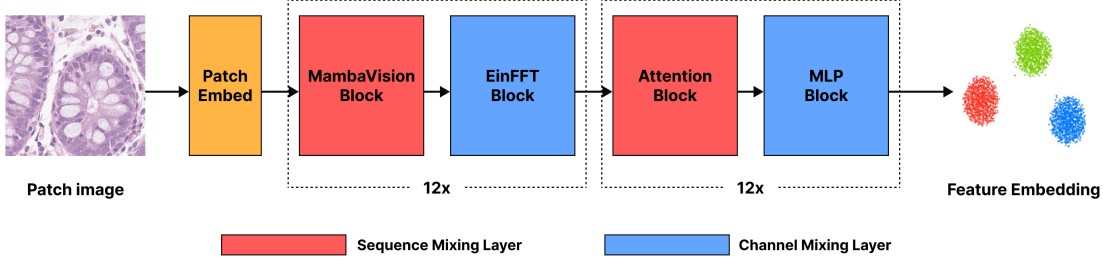

Figure 1: Architecture of MVHybrid showing the hybrid state space-ViT blocks with interleaved sequence and channel mixing layers. Sequence mixing layers are in red and channel mixing layers are in blue.

different combinations of sequence and channel mixing layers. All models have an equal embedding dimension of 384 and follow the default "Small" configuration. $ViT_{12}$ and $ViT_{24}$ are included to be the original ViT-Small baseline (12 layers) and a one-to-one comparison with other models (24 layers), respectively. Furthermore, it was empirically found that all Mamba-based sequence mixers are incompatible with MLP channel mixers due to unstable pretraining (possibly due to positive real eigenvalues) in DINOv2, and therefore we use EinFFT channel mixing blocks instead of Multi-layer Perceptron (MLP) for increased pretraining stability as reported in Patro and Agneeswaran (2024) (more details about pretrained models are in Appendix B). After pretraining these models, the teacher is used as a pretrained encoder to generate meaningful feature embeddings when processing input WSI patches during inference.

|  | $ViM_{EinFFT}$ | $Hydra_{EinFFT}$ | $ViT_{12}$ | $ViT_{24}$ | $Hydra_{Hybrid}$ | $MV_{Hybrid}$ |
|---|---|---|---|---|---|---|
| **Sequence Mixer** | ViM | Hydra | Attention | Attention | Hydra&Attention | MV&Attention |
| **Channel Mixer** | EinFFT | EinFFT | MLP | MLP | EinFFT&MLP | EinFFT&MLP |

Table 1: Pretrained Models and Their Components. The naming convention is sequence mixer followed by a subscript of the channel mixer.

## 2.2 Datasets and Experiments

For all experiments, we used publicly available histopathology datasets from CRC Hematoxylin & Eosin (H&E) WSIs. All WSI-based datasets followed the identical preprocessing method of background removal and morphological closings (Lu et al., 2021) to patch the WSI into 256 x 256 image resolution patches that only contain relevant tissue. All other patch-based evaluation datasets were resized to 256 x 256. For pretraining all of the models in Table 1, we used the same CRC pretraining dataset curated by randomly selecting WSIs in a class-stratified (normal, benign, malignant) manner from the HunCRC (Ármin Pataki

et al., 2022) and IMP-CRS-2024 (Neto et al., 2024) dataset. Full details of pretraining data curation and experiments are in Appendix C.1.

**Biomarker Prediction.** For evaluating the models on biomarker prediction, we used the paired ST-H&E data from HEST, which contain a mix of 10X (10x Genomics, 2025) Visium, VisiumHD, and Xenium data. We use Jaume et al. (2024)'s official k-fold cross-validation (CV) split from HEST-Benchmark, but combine the HEST-COAD and READ dataset in a patient stratified manner. For the gene set of HEST-Benchmark, we use the given top 50 highly variable genes (HVG) and their normalized expressions. HEST-Benchmark only contains eight samples from four patients, so we curate another "HEST-Extended" dataset that consists of 54 samples from eight study sources that are not part of HEST-Benchmark, further extracted from the HEST dataset. HEST-Extended data are used for training in two different ways: 1) Random 10-fold CV where all data is randomly split between train and test regardless of study source, 2) LOSO (Leave-One-Study-Out) where samples from one study are left as a test set and all other samples are used in training. Observing each of the pretrained model's performance and its drop between random and LOSO signifies the overall quality and the robustness of each model's embeddings, respectively. For the gene set of HEST-Extended, we extract top 200 HVGs using the same process as Jaume et al. (2024), but also extract top 200 high mean HVGs (HMHVGs) similar to Zhu et al. (2025) which show high mean expression as well. HVGs capture genes with high expression variance across samples regardless of their baseline levels, identifying biological heterogeneity and functional diversity, while HMHVGs select genes that are both abundantly expressed and highly variable, revealing how core biological processes are differentially regulated across tissue regions. We ensure to only include genes that are present in all samples so that all patches have a paired gene expression value.

HEST-Benchmark and HEST-Extended are both evaluated by training a downstream Ridge regression model on the extracted patch embeddings from each pretrained model to predict the gene expression values of the HVG and HMHVG sets. The trained regression model is then evaluated by inferring the gene expression of the patches from the test set. The ground truth value is then compared to the predicted gene expression via the following metrics: Pearson correlation coefficient (PCC) (for top-10 and all genes), mean absolute error (MAE), and mean squared error (MSE). Full details of data curation and experiments for biomarker prediction and for other downstream tasks like classification, survival prediction and patch retrieval are in Appendix C.2 and C.3, respectively.

## 3 Results and Discussion

We first show evaluation results for biomarker prediction, and briefly mention key results for other downstream tasks. We use Ridge regression as the sole downstream evaluation model for biomarker prediction to maintain evaluation consistency with the classification setting of VFMs, such as linear probing. Ridge regression serves as a simple yet effective method to assess the quality of the extracted embeddings—specifically their linear separability—without the confounding effects of more complex regression models. Our results demonstrate that $MV_{Hybrid}$ exhibits superior biomarker prediction ability and robustness compared to all other models including ViT. HEST-Benchmark results (Table 2) show $MV_{Hybrid}$ achieves the highest correlations (PCC, PCC-10) and lowest errors (MAE, MSE) across all mod-

| Model | PCC | PCC-10 | MAE | MSE |
|---|---|---|---|---|
| $ViM_{EinFFT}$ | 0.397±0.065 | 0.685±0.069 | 1.896±0.332 | 5.956±1.985 |
| $Hydra_{EinFFT}$ | 0.404±0.064 | 0.692±0.067 | 1.879±0.270 | 5.781±1.674 |
| $ViT_{12}$ | 0.415±0.055 | 0.720±0.097 | 1.807±0.355 | 5.392±2.064 |
| $ViT_{24}$ | 0.365±0.042 | 0.664±0.080 | 1.869±0.285 | 5.822±1.834 |
| $Hydra_{Hybrid}$ | 0.415±0.069 | 0.688±0.082 | 1.824±0.386 | 5.618±2.157 |
| **$MV_{Hybrid}$** | **0.460±0.082** | **0.747±0.082** | **1.748±0.265** | **5.011±1.478** |

Table 2: HEST-Benchmark Results.

els. PCC measures how well the regression model captures the linear relationship between predicted and actual gene expression values, while MAE/MSE quantify the magnitude of prediction errors in absolute terms—excelling in both demonstrates that $MV_{Hybrid}$ generates superior embeddings that accurately predict both relative expression patterns and actual expression values. HEST-Extended results (Tables 3 and 4) show that in LOSO evaluation, $MV_{Hybrid}$ ranks first across both gene sets with PCC scores of 0.138 (HVG) and 0.212 (HMHVG), outperforming the best-performing ViT by 42% (HVG) and 71% (HMHVG). In addition, $MV_{Hybrid}$ is the most robust model as it achieves the lowest PCC decrease (35.5% HVG, 46.0% HMHVG) and PCC-10 decrease, and the lowest MSE (48.2% HVG, 29.7% HMHVG) and MAE (25.9% HVG, 10.2% HMHVG) increase, suggesting $MV_{Hybrid}$ captures more biological features than site-specific features. This superior performance and robustness is only partly due to $MV_{Hybrid}$'s bias toward lower frequencies, as other Mamba-based models achieve lower performance. Therefore, MV's vision-specific design of including regular convolution layers in both SSM and skip connection paths seems to help more than the bidirectional processing from Hydra as $Hydra_{Hybrid}$ shows inferior results. Also, the hybrid nature of $MV_{Hybrid}$ seems to allow MV and ViT layers to capture fundamentally different features, as pure SSM-based models like $ViM_{EinFFT}$ and $Hydra_{EinFFT}$ exhibit weaker performance. Interestingly, while HMHVGs show higher correlation values than HVGs,

| Metric | Eval | $ViM_{EinFFT}$ | $Hydra_{EinFFT}$ | $ViT_{12}$ | $ViT_{24}$ | $Hydra_{Hybrid}$ | **$MV_{Hybrid}$** |
|---|---|---|---|---|---|---|---|
| **PCC** | Random | 0.154±0.119 | 0.218±0.130 | 0.210±0.146 | 0.176±0.115 | 0.191±0.104 | 0.214±0.122 |
| | LOSO | 0.083±0.086 | 0.116±0.097 | 0.097±0.108 | 0.089±0.091 | 0.094±0.080 | **0.138±0.102** |
| | Decrease (%) | 46.4 | 46.8 | 53.7 | 49.5 | 51.1 | **35.5** |
| **PCC-10** | Random | 0.526±0.135 | 0.570±0.136 | 0.555±0.143 | 0.540±0.134 | 0.540±0.137 | 0.564±0.129 |
| | LOSO | 0.314±0.132 | 0.334±0.152 | 0.349±0.174 | 0.337±0.123 | 0.335±0.143 | **0.386±0.175** |
| | Decrease (%) | 40.3 | 41.5 | 37.2 | 37.6 | 38.0 | **31.5** |
| **MSE** | Random | 0.634±0.333 | 0.600±0.288 | 0.593±0.240 | 0.612±0.300 | 0.594±0.274 | 0.594±0.283 |
| | LOSO | 1.107±0.803 | 1.036±0.717 | 1.003±0.642 | 0.975±0.741 | 0.954±0.666 | **0.881±0.671** |
| | Increase (%) | 74.6 | 72.8 | 69.2 | 59.2 | 60.6 | **48.2** |
| **MAE** | Random | 0.495±0.133 | 0.491±0.121 | 0.488±0.101 | 0.488±0.120 | 0.486±0.113 | 0.488±0.120 |
| | LOSO | 0.706±0.319 | 0.686±0.297 | 0.674±0.265 | 0.644±0.299 | 0.648±0.277 | **0.614±0.281** |
| | Increase (%) | 42.5 | 39.7 | 38.1 | 31.9 | 33.4 | **25.9** |

Table 3: HEST-Extended HVG (n = 200) Results: Random vs LOSO

they exhibit higher MAE/MSE values across all models. This arises because HVGs have high variance but low mean expression, resulting in smaller errors despite lower correlations while HMHVGs have high mean expression values as well, leading to larger errors even when correlations are stronger. This highlights that both metrics are necessary—correlation captures the model's ability to predict relative expression patterns, while MAE/MSE reflect prediction accuracy in absolute expression units. $MV_{Hybrid}$ shows equal or slightly better

| Metric | Eval | ViM$_\text{EinFFT}$ | Hydra$_\text{EinFFT}$ | ViT$_{12}$ | ViT$_{24}$ | Hydra$_\text{Hybrid}$ | MV$_\text{Hybrid}$ |
|---|---|---|---|---|---|---|---|
| **PCC** | Random | 0.309±0.179 | 0.400±0.184 | 0.373±0.212 | 0.334±0.167 | 0.367±0.155 | 0.393±0.162 |
| | LOSO | 0.124±0.183 | 0.154±0.144 | 0.110±0.203 | 0.124±0.179 | 0.122±0.148 | **0.212±0.166** |
| | Decrease (%) | 59.8 | 61.6 | 70.6 | 62.8 | 66.7 | **46.0** |
| **PCC-10** | Random | 0.570±0.129 | 0.625±0.128 | 0.605±0.142 | 0.583±0.117 | 0.603±0.119 | 0.620±0.116 |
| | LOSO | 0.356±0.171 | 0.378±0.164 | 0.377±0.182 | 0.394±0.146 | 0.357±0.169 | **0.454±0.168** |
| | Decrease (%) | 37.5 | 39.5 | 37.6 | 32.5 | 40.8 | **26.8** |
| **MSE** | Random | 4.837±1.758 | 4.555±1.487 | 4.581±1.344 | 4.707±1.638 | 4.542±1.409 | 4.542±1.454 |
| | LOSO | 8.060±5.736 | 7.065±5.057 | 7.123±5.225 | 6.865±5.078 | 6.727±4.382 | **5.889±4.161** |
| | Increase (%) | 66.7 | 55.1 | 55.5 | 45.9 | 48.1 | **29.7** |
| **MAE** | Random | 1.806±0.373 | 1.778±0.346 | 1.780±0.293 | 1.789±0.346 | 1.772±0.324 | 1.776±0.337 |
| | LOSO | 2.290±1.098 | 2.164±0.950 | 2.174±0.923 | 2.098±0.986 | 2.102±0.904 | **1.957±0.853** |
| | Increase (%) | 26.8 | 21.7 | 22.1 | 17.2 | 18.6 | **10.2** |

Table 4: HEST-Extended HMHVG (n = 200) Results: Random vs LOSO

performance in three different downstream tasks: classification, patch retrieval, and survival prediction (detailed results and discussion are in Appendix D). We believe that this is also due to MV's favorable design (regular convolution), hybrid ViT (attention is proven for its strong representations), and its low-frequency bias—shown in the eigenvalue distribution analysis in Figure 2 of Appendix E. Figure 2 shows that all pretrained Mamba variants maintain negative real eigenvalues, with MV$_\text{Hybrid}$'s broader eigenvalue distribution creating cascaded low-pass filters with diverse cutoff frequencies (at $\omega_c = |\lambda_j|$), which according to the theoretical analysis in Section 1.1 provides progressively stronger attenuation at higher frequencies while preserving a richer set of low-frequency features.

## 4 Conclusion

With the broad distribution of negative real eigenvalues resulting in low-frequency bias of MV layers in MV$_\text{Hybrid}$ combined with its vision-centric and hybrid ViT design, we show that MV$_\text{Hybrid}$ outperforms ViTs in biomarker prediction performance and robustness when pretrained and evaluated on the same dataset. We show that tailoring the backbone architecture of pathology VFMs is effective, especially as current VFMs are shown to be unrobust despite being trained on large-scale WSI datasets. We empirically show that biomarker prediction performance is partly correlated with the backbone's bias for low-frequency features. We leave performing ablation studies for each sequence and channel mixers of MV$_\text{Hybrid}$ to analyze how individual modifications impact performance to future work. Furthermore, more extensive validation on public and clinical paired ST-H&E data is needed. Despite these limitations, MV$_\text{Hybrid}$'s superior performance in the critical task of biomarker prediction and competitive or better performance across other downstream tasks positions it as a compelling architecture for future pathology VFMs.

### Acknowledgments and Disclosure of Funding

This research was supported by a grant from the Korea Health Technology R&D Project through the Korea Health Industry Development Institute (KHIDI), funded by the Ministry of Health & Welfare, Republic of Korea (grant number: RS-2021-KH113146).

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

## Appendix A. Derivation of Enhanced Low-Frequency Bias for Negative Real Eigenvalues

In this appendix, we summarize the derivation of Yu et al. (2025) in A.1 and A.2 first to show in A.3 that the same derivation can be applied to prove that negative real eigenvalues have an even stronger bias for low-frequency. A.4 analyzes the low-frequency bias of complex and negative real eigenvalues and how they differ.

### A.1 Total Variation of Transfer Function

Starting from the continuous-time SSM equations (1) and (2), we derive the transfer function $G(is)$ as shown in equation (3). While practical implementations use discretized forms with discretization step $\Delta$, where $\bar{A} = \exp(\Delta A)$, the frequency analysis remains valid as the discretization preserves the eigenvalue structure of $A$.

The total variation of the transfer function $G(is)$ over a frequency interval $[a, b]$ quantifies the cumulative change in the frequency response as shown in Equation (4) of the main text:

$$V_a^b(G) = \int_a^b \left| \frac{dG(is)}{ds} \right| ds \tag{7}$$

Given the transfer function in partial fraction form:

$$G(is) = \sum_{j=1}^N \frac{c_j}{is - a_j} + D \tag{8}$$

The derivative with respect to frequency $s$ is:

$$\frac{dG(is)}{ds} = \sum_{j=1}^N \frac{-ic_j}{(is - a_j)^2} \tag{9}$$

### A.2 Case 1: Complex Eigenvalues

For complex eigenvalues, we summarize Yu et al. (2025)'s derivation as below to compare with negative real eigenvalues (shown in A.3 below). For complex eigenvalues $a_j = v_j + iw_j$ where $v_j < 0$ (stability condition):

$$\frac{dG(is)}{ds} = \sum_{j=1}^N \frac{-ic_j}{(is - v_j - iw_j)^2} \tag{10}$$

The magnitude of the denominator is:

$$|is - a_j| = |is - v_j - iw_j| = \sqrt{v_j^2 + (s - w_j)^2} \tag{11}$$

Therefore:

$$\left| \frac{dG(is)}{ds} \right| = \sum_{j=1}^N \frac{|c_j|}{v_j^2 + (s - w_j)^2} \tag{12}$$

For high frequencies where $s \gg \max(|w_j|)$, the dominant term is $(s - w_j)^2$, yielding:

$$\left| \frac{dG(is)}{ds} \right| \approx \sum_{j=1}^{N} \frac{|c_j|}{(s - w_j)^2} \tag{13}$$

The high-frequency total variation is:

$$V_{\omega_0}^{\infty}(G) = \sum_{j=1}^{N} |c_j| \int_{\omega_0}^{\infty} \frac{1}{(s - w_j)^2} ds \tag{14}$$

$$= \sum_{j=1}^{N} |c_j| \left[ -\frac{1}{s - w_j} \right]_{\omega_0}^{\infty} \tag{15}$$

$$= \sum_{j=1}^{N} \frac{|c_j|}{\omega_0 - w_j} \tag{16}$$

This gives us the bound (assuming $\omega_0 > \max(|w_j|)$ for convergence):

$$V_{\omega_0}^{\infty}(G) \leq \sum_{j=1}^{N} \frac{|c_j|}{|w_j - \omega_0|} \tag{17}$$

### A.3 Case 2: Negative Real Eigenvalues

For negative real eigenvalues $a_j = -|\lambda_j|$ where $\lambda_j > 0$:

$$|is - a_j| = |is + |\lambda_j|| = \sqrt{|\lambda_j|^2 + s^2} \tag{18}$$

This yields:

$$\left| \frac{dG(is)}{ds} \right| = \sum_{j=1}^{N} \frac{|c_j|}{(|\lambda_j|^2 + s^2)} \tag{19}$$

The high-frequency total variation becomes:

$$V_{\omega_0}^{\infty}(G) = \sum_{j=1}^{N} |c_j| \int_{\omega_0}^{\infty} \frac{1}{|\lambda_j|^2 + s^2} ds \tag{20}$$

**Step-by-Step Derivation:** Using the standard integral identity

$$\int \frac{1}{a^2 + x^2} dx = \frac{1}{a} \arctan\left(\frac{x}{a}\right) + C \tag{21}$$

we obtain:

$$\int_{\omega_0}^{\infty} \frac{1}{|\lambda_j|^2 + s^2} ds = \frac{1}{|\lambda_j|} \left[ \arctan\left(\frac{s}{|\lambda_j|}\right) \right]_{\omega_0}^{\infty} \tag{22}$$

$$= \frac{1}{|\lambda_j|} \left[ \frac{\pi}{2} - \arctan\left(\frac{\omega_0}{|\lambda_j|}\right) \right] \tag{23}$$

For high frequencies where $\omega_0 \gg |\lambda_j|$, we use the large-$x$ approximation:

$$\arctan(x) \approx \frac{\pi}{2} - \frac{1}{x} \quad \text{for } x \gg 1 \tag{24}$$

Letting $x = \frac{\omega_0}{|\lambda_j|}$, we have:

$$\arctan\left(\frac{\omega_0}{|\lambda_j|}\right) \approx \frac{\pi}{2} - \frac{|\lambda_j|}{\omega_0} \tag{25}$$

Substituting into the integral:

$$\int_{\omega_0}^{\infty} \frac{1}{|\lambda_j|^2 + s^2} ds = \frac{1}{|\lambda_j|}\left[\frac{\pi}{2} - \arctan\left(\frac{\omega_0}{|\lambda_j|}\right)\right] \tag{26}$$

$$\approx \frac{1}{|\lambda_j|}\left[\frac{\pi}{2} - \left(\frac{\pi}{2} - \frac{|\lambda_j|}{\omega_0}\right)\right] \tag{27}$$

$$= \frac{1}{|\lambda_j|} \cdot \frac{|\lambda_j|}{\omega_0} = \frac{1}{\omega_0} \tag{28}$$

This shows that for $\omega_0 \gg |\lambda_j|$, the integral decays as $1/\omega_0$.

**High-Frequency Approximation:** The arctangent approximation shows $O(1/\omega_0)$ decay. We derive a refined approximation capturing $|\lambda_j|$'s role.

Starting from the exact integral derived above:

$$\int_{\omega_0}^{\infty} \frac{1}{|\lambda_j|^2 + s^2} ds = \frac{1}{|\lambda_j|}\left[\frac{\pi}{2} - \arctan\left(\frac{\omega_0}{|\lambda_j|}\right)\right] \tag{29}$$

For high frequencies where $\omega_0 \gg |\lambda_j|$, we can analyze the asymptotic behavior. Using the expansion $\arctan(x) \approx \frac{\pi}{2} - \frac{1}{x} + O(1/x^3)$ for large $x$:

$$\int_{\omega_0}^{\infty} \frac{1}{|\lambda_j|^2 + s^2} ds \approx \frac{1}{|\lambda_j|} \cdot \frac{|\lambda_j|}{\omega_0} = \frac{1}{\omega_0} \tag{30}$$

A more insightful approximation that captures both the asymptotic behavior and the transition region is:

$$\int_{\omega_0}^{\infty} \frac{1}{|\lambda_j|^2 + s^2} ds \sim \frac{1}{\sqrt{|\lambda_j|^2 + \omega_0^2}} \tag{31}$$

This approximation is particularly useful because:

1. For $\omega_0 \gg |\lambda_j|$: $\frac{1}{\sqrt{|\lambda_j|^2 + \omega_0^2}} \approx \frac{1}{\omega_0}$, recovering the correct asymptotic behavior

2. For $\omega_0 \sim |\lambda_j|$: It captures the transition where the eigenvalue $|\lambda_j|$ significantly affects the response

3. The form $\frac{1}{\sqrt{|\lambda_j|^2 + \omega_0^2}}$ represents the magnitude response of a first-order low-pass filter with cutoff at $|\lambda_j|$

Therefore, the high-frequency total variation can be approximated as:

$$V_{\omega_0}^\infty(G) \sim \sum_{j=1}^N \frac{|c_j|}{\sqrt{|\lambda_j|^2 + \omega_0^2}} \tag{32}$$

This approximation reveals that each negative real eigenvalue $|\lambda_j|$ acts as a low-pass filter with cutoff frequency $\omega_c = |\lambda_j|$, and the overall frequency response is determined by the superposition of these filters.

## A.4 Comparison and Enhanced Low-Frequency Bias

The key insight emerges from comparing the decay rates:

- **Complex eigenvalues**: $V_{\omega_0}^\infty(G) \sim \sum_j \frac{|c_j|}{\omega_0 - w_j}$ (linear decay)

- **Negative real eigenvalues**: $V_{\omega_0}^\infty(G) \sim \sum_j \frac{|c_j|}{\omega_0}$ (uniform decay)

For large $\omega_0$:

$$\frac{1}{\omega_0} < \frac{1}{\omega_0 - w_j} \quad \text{for any finite } w_j < \omega_0 \tag{33}$$

This demonstrates that negative real eigenvalues provide:

1. **Faster high-frequency decay**: The $\frac{1}{\omega_0}$ decay is uniformly faster than $\frac{1}{\omega_0 - w_j}$

2. **Uniform frequency response**: All eigenvalues contribute equally to the decay, creating a smooth roll-off

3. **Sharp cutoff characteristic**: With $\lambda_j = [1, 2, 3, \ldots, N]$, the system acts as a cascade of low-pass filters with cutoffs at integer frequencies

The magnitude response for negative real eigenvalues:

$$|G(i\omega)| = \left| \sum_{j=1}^N \frac{c_j}{i\omega + |\lambda_j|} \right| \approx \begin{cases} \sum_j \frac{|c_j|}{|\lambda_j|} & \text{if } \omega \ll \min(|\lambda_j|) \\ \sum_j \frac{|c_j|}{\omega} & \text{if } \omega \gg \max(|\lambda_j|) \end{cases} \tag{34}$$

This creates a uniform -20 dB/decade roll-off beyond the maximum eigenvalue, effectively implementing a higher-order low-pass filter ideal for preserving low-frequency biological patterns while suppressing high-frequency noise in pathology images.

The specific initialization schemes employed by our models further enhance this effect:

- **MambaVision/ViM** ($\lambda_j = [1, 2, 3, \ldots, N]$): Creates cascaded low-pass filters with cutoff frequencies at $\omega_c = 1, 2, 3, \ldots, N$, resulting in progressively stronger attenuation at higher frequencies.

- **Hydra** ($\lambda_j = [1, 1, \ldots, 1]$): Creates $N$ identical low-pass filters with cutoff at $\omega_c = 1$, providing consistent attenuation across all channels.

This initialization scheme impacts the eigenvalue profiles of the pretrained models, which is shown in Figure 2 of Appendix E.

## Appendix B. Descriptions of All Pretrained Models

In this appendix, we include the detailed descriptions of all pretrained models listed in Table 1. Table 5 below provides comprehensive details about all models used in our experiments, including their architectural components, computational requirements, and performance characteristics.

| Model | Sequence Mixer | Channel Mixer | # Params (M) | GFLOPs (256) | GFLOPs (512) | Throughput (256) (img/s) | Throughput (512) (img/s) | Ratio |
|---|---|---|---|---|---|---|---|---|
| ViM$_\text{EinFFT}$ | ViM | EinFFT | 29.0 | 8.1 | 32.3 | 502 | 115 | 4.365 |
| Hydra$_\text{EinFFT}$ | Hydra | EinFFT | 28.2 | 8.1 | 32.4 | 494 | 114 | 4.333 |
| ViT$_{12}$ | Attention | MLP | 21.7 | 6.2 | 31.8 | 3,346 | 435 | 7.692 |
| ViT$_{24}$ | Attention | MLP | 43.0 | 12.2 | 63.3 | 1,694 | 219 | 7.735 |
| Hydra$_\text{Hybrid}$ | Hydra/Attention | EinFFT/MLP | 35.5 | 10.2 | 47.8 | 775 | 138 | 5.616 |
| **MV$_\text{Hybrid}$** | MV/Attention | EinFFT/MLP | 30.9 | 8.4 | 33.4 | 1,119 | 231 | 4.844 |

Table 5: Table of All Pretrained Models and their Efficiency Profiles. The naming convention follows sequence mixer followed by channel mixer in subscript. Hybrid signifies a hybrid model where the second half of the model is a vanilla ViT. Throughput is measured on a NVIDIA RTX 4090 GPU, in images per second (img/s). 256 and 512 signify 256 x 256 and 512 x 512 patch size.

We first choose to train ViM$_\text{EinFFT}$ for a baseline Mamba performance and train MV$_\text{Hybrid}$ to follow MV's performance. Then, we make the same modifications to Hydra that we made for MV$_\text{Hybrid}$ to train Hydra$_\text{Hybrid}$ and Hydra$_\text{EinFFT}$. As shown in the number of parameters and GFLOPs above, all pure Mamba and hybrid models have a lower number of parameters and GFLOPs compared to ViT$_{24}$. While ViT$_{24}$ has higher throughput, MV$_\text{Hybrid}$ is a close second and is highest of all other models. Furthermore, MV$_\text{Hybrid}$ enjoys favorable scaling properties as the throughput ratio is near-linear compared to that of ViT. It also beats ViT$_{24}$ in throughput for image sizes of 512 x 512 (231 vs 219 img/s), showing its potential for application in larger image sizes as it has higher throughput with lower GFLOPs and number of parameters. Since MV$_\text{Hybrid}$ is a mix of MV and ViT, it is impressive that the scaling remains near-linear.

## Appendix C. Details of Dataset and Experiments

In this appendix, we detail the data curation and experiments for pretraining in C.1, data curation and experiments for biomarker prediction evaluation in C.2, and data curation and experiments for all other evaluation tasks (classification, patch retrieval, survival prediction) in C.3.

### C.1 Pretraining Data Curation and Experiments

For data curation for the pretraining dataset, we first downloaded the HunCRC and IMP-CRS-2024 datasets that contain 200 and 5,333 WSIs, respectively. Both contain three classes of normal, benign, and malignant and are scanned at 40x magnification. 25 and 15 WSIs were randomly selected in a class-stratified manner from IMP-CRS-2024 and HunCRC, respectively. After preprocessing, a total of 756,000 tissue patches were used to pretrain

all the models via DINOv2 for 200 epochs using a learning rate of 2.5e-3 and batch size of 1,536. The pretraining dataset is not used for evaluation, and it is made sure that there are no overlaps between the training and evaluation datasets.

### C.2 Data Curation and Experiments for Biomarker Prediction Evaluation

For data curation and experiments for biomarker prediction evaluation, below are the descriptions for HEST-Benchmark and HEST-Extended, which are both part of the HEST-1k dataset, but curated differently and contain no overlaps.

**HEST-Benchmark**: HEST as a total contains 1,229 paired spatial transcriptomics (ST) and WSIs from 26 organs. We only utilize colon and rectum benchmark datasets, consisting of eight WSI-ST pairs from four patients. From that, we use the given top 50 most variable gene expression values and train a Ridge regression model to predict the gene expressions by only using the extracted feature embeddings from the models. We use patient-wise cross-validation, resulting in 4-folds of train/test split with a 3:1 split. Pearson correlation is used as an evaluation metric.

**HEST-Extended**: We use all the HEST data that is not part of HEST-Benchmark to collect a total of 56 samples from COAD, READ, and COADREAD categories, where these 56 samples come from 8 different study sources. Two samples were eliminated because their number of genes were significantly less than the other samples, preventing the calculation of HVGs and HMHVGs (gene overlap must be calculated for all samples first) to leave 54 samples. As mentioned in the main text, a random 10-fold CV and an 8-fold LOSO dataset was curated. Top 200 HVGs were first measured with the highly variable genes function of *scanpy* (Wolf et al., 2018) with log1p normalization. HMHVGs were then calculated by listing all genes with high mean in one list, and listing the HVGs in another list and finding the overlaps with one another to form top 200 HMHVGs.

### C.3 Data Curation and Experiments for All Other Evaluation Tasks

For all other downstream evaluation tasks, below is the description of the dataset curation and the experiments performed. Recall that all of these downstream evaluation tasks, like biomarker prediction, are all trained on the extracted patch embeddings of the pretrained VFMs. If not specifically mentioned below, the default given train-test split was used.

**TCGA-CRC-MSI (Binary classification)**: This dataset contains a total of 535 WSIs from The Cancer Genome Atlas (The Cancer Genome Atlas Research Network, 2006). Only WSIs with microsatellite instability (MSI) information were used, which were curated by filtering the TCGA-COAD and TCGA-READ datasets by their MSI Mantis Score (Kautto et al., 2017) on cBioPortal (Cerami et al., 2012). Filtering by the default threshold of $\leq 0.4$ and $> 0.6$ returned 468 MSS (microsatellite stable) and 67 MSI-high WSIs, respectively. Due to this class imbalance, we created ten different balanced folds with MSI-high fixed, and randomly sampling an equal number of MSS cases. A Clustering-constrained Attention Multiple Instance Learning (CLAM) model was trained for 200 epochs and Area Under the Receiver Operating Characteristic (AUROC) and mean accuracy (mAcc) were used as evaluation metrics.

**MHIST (Wei et al., 2021) (Binary classification)**: This dataset consists of 3,152 patch images sized 224 x 224 at 5x magnification. The binary classes are hyperplastic

polyps (HP) and sessile serrated adenomas (SSA). A logistic regression model was trained for linear probing, evaluated on AUROC and balanced accuracy. The K-nearest neighbor (KNN) framework was used to cluster the feature embeddings for KNN probing ($K = 20$) and few-shot (SimpleShot) (Wang et al., 2019) framework was utilized to evaluate the model's feature representations. $K = 4$ samples for each class were used to generate a class prototype and all other samples are tested via nearest L2 distance ($n = 1000$). Both unsupervised models were evaluated using weighted F1 (WF1) and balanced accuracy (BAcc).

**UniToPatho (Barbano et al., 2021) (6-class classification)**: This dataset comprises 9,536 patch images at 20x magnification for polyp classification and adenoma grading. Only the subset containing 8,669 images of size 1,812 x 1,812 pixels was used. The exact same downstream training and evaluation metrics were utilized as that of MHIST.

**NCT-CRC-100K (Kather et al., 2018) (9-class zero-shot patch retrieval)**: This dataset includes 100,000 patch images from nine tissue classes at 20x magnification. The models were evaluated with zero-shot patch retrieval where test embeddings query against training embeddings. Features were normalized and searched using FAISS IndexFlatL2 (Douze et al., 2024). Performance was measured with accuracy: Acc@K ($K \in \{1,3,5\}$) and MVAcc@5 (Majority voting accuracy). The former considers retrieval successful if any of top-K patches match the query label, the latter requires the query to align with the majority vote from the top-5 retrieved patches.

**TCGA-CRC (Survival prediction)**: This dataset is identical to TCGA-CRC-MSI but is unfiltered. We follow the default 5-fold train-test split of PANTHER (Song et al., 2024) and train a survival prediction model by utilizing extracted feature embeddings to train an unsupervised Gaussian Mixture Model (GMM). This is a prototype-based learning method that is more sensitive to the feature embeddings compared to supervised models. Commonly used concordance metric (c-index) was used for evaluation.

## Appendix D. Results for Other Tasks: Classification, Patch Retrieval, and Survival Prediction

In this appendix, we include the results of three classification tasks, patch retrieval, and survival prediction in Tables 6 and 7 below.

The results in Table 6 show the models' performance on the three different classification tasks. Notably, MVHybrid outperforms both ViTs across all metrics and achieves the best performance in all metrics except for three, where it is a close second. MVHybrid excels at classifying MSI/MSS biomarkers, which is a hallmark prognostic biomarker in CRC. Unlike most classification evaluation tasks which are morphology-based, MSI and MSS status are molecular-based and cannot be clearly distinguished via morphology in WSIs. We believe that MVHybrid's superior performance over both ViTs on molecular tasks is also particularly due to its low-frequency bias as HydraHybrid also shows high performance.

Furthermore, MVHybrid shows superior performance on the MHIST and UniToPatho datasets (morphology-based classification tasks), outperforming both ViTs , signifying that its unique design is also effective in creating strong representations of tissue morphology. This is confirmed in the linear and KNN probing performance, as it directly measures the representation quality of the extracted embeddings (linear probing evaluates linear separability while KNN probing evaluates the clusters in an unsupervised and nonparametric

| Dataset | Method | Metric | $\text{ViM}_{\text{EinFFT}}$ | $\text{Hydra}_{\text{EinFFT}}$ | $\text{ViT}_{12}$ | $\text{ViT}_{24}$ | $\text{Hydra}_{\text{Hybrid}}$ | $\textbf{MV}_{\textbf{Hybrid}}$ |
|---|---|---|---|---|---|---|---|---|
| TCGA-CRC-MSI | CLAM | AUC | 0.730±0.089 | **0.772±0.103** | 0.706±0.116 | 0.746±0.134 | 0.763±0.103 | 0.765±0.090 |
| | | mAcc | 0.668±0.072 | 0.707±0.098 | 0.657±0.139 | 0.696±0.086 | 0.718±0.102 | **0.750±0.073** |
| MHIST | Linear Probing | AUC | 0.793 | 0.837 | 0.831 | 0.804 | 0.855 | **0.863** |
| | | BAcc | 0.689 | 0.720 | 0.713 | 0.705 | 0.758 | **0.768** |
| | KNN Probing | WF1 | 0.648 | 0.687 | 0.700 | 0.667 | 0.699 | **0.743** |
| | | BAcc | 0.605 | 0.643 | 0.664 | 0.624 | 0.656 | **0.703** |
| | Few-shot | WF1 | 0.535±0.056 | 0.560±0.051 | 0.553±0.058 | 0.542±0.057 | 0.562±0.047 | **0.575±0.062** |
| | | BAcc | 0.543±0.042 | 0.573±0.046 | 0.578±0.054 | 0.560±0.051 | 0.578±0.049 | **0.595±0.058** |
| UniToPatho | Linear Probing | mAUC | 0.791 | 0.806 | 0.801 | 0.789 | 0.802 | **0.820** |
| | | BAcc | 0.396 | 0.416 | 0.416 | 0.403 | 0.405 | **0.463** |
| | KNN Probing | WF1 | 0.426 | **0.451** | 0.438 | 0.434 | 0.435 | 0.444 |
| | | BAcc | 0.328 | 0.334 | 0.333 | 0.365 | 0.329 | **0.373** |
| | Few-shot | WF1 | 0.290 ± 0.047 | 0.310 ± 0.050 | 0.306 ± 0.053 | **0.319 ± 0.058** | 0.299 ± 0.048 | 0.310 ± 0.051 |
| | | BAcc | 0.306 ± 0.038 | 0.316 ± 0.039 | 0.321 ± 0.047 | 0.325 ± 0.051 | 0.308 ± 0.037 | **0.333 ± 0.042** |

Table 6: Classification Results

| Dataset | Metric | $\text{ViM}_{\text{EinFFT}}$ | $\text{Hydra}_{\text{EinFFT}}$ | $\text{ViT}_{12}$ | $\text{ViT}_{24}$ | $\text{Hydra}_{\text{Hybrid}}$ | $\textbf{MV}_{\textbf{Hybrid}}$ |
|---|---|---|---|---|---|---|---|
| NCT-CRC-100K | Recall@1 | 0.762 | 0.763 | 0.674 | 0.648 | 0.774 | **0.789** |
| | Recall@3 | 0.870 | 0.847 | 0.783 | 0.763 | 0.877 | **0.880** |
| | Recall@5 | 0.898 | 0.880 | 0.825 | 0.808 | 0.907 | **0.911** |
| | MVAcc@5 | 0.810 | 0.780 | 0.714 | 0.693 | 0.808 | **0.825** |
| TCGA-CRC | Mean c-index | 0.601±0.056 | **0.677±0.074** | 0.623±0.081 | 0.620±0.132 | 0.651±0.071 | 0.658±0.076 |

Table 7: Patch Retrieval and Survival Prediction Results

way). Few-shot learning, which is also unsupervised and nonparametric, creates a class prototype for each class using K samples and performs nearest centroid classification for the rest of the dataset for testing. $\text{MV}_{\text{Hybrid}}$'s compelling performance shows its robustness to unseen datasets within the same dataset distribution. $\text{MV}_{\text{Hybrid}}$'s strong performance in UniToPatho (3x magnification after resizing) also shows its robustness to low magnification images as well.

The results in Table 7 exhibit the models' performance on patch retrieval and biomarker or survival prediction. WSI retrieval is clinically important in diagnosis and medical research. While different, patch retrieval still can be viewed as a subproblem that addresses similar technical challenges. In zero-shot patch retrieval, $\text{MV}_{\text{Hybrid}}$ shows its superior ability to find visually similar images as it outperforms both ViTs on all four metrics. Lastly, survival prediction is also important in the clinic for cancer prognostics and is a unique task because it can't be classified into a morphological or molecular-based task as multiple features of the image can contribute to survival prediction. In survival prediction, $\text{MV}_{\text{Hybrid}}$ also outperforms both ViTs.

Overall, while we report that $\text{MV}_{\text{Hybrid}}$ outperforms ViTs in almost all metrics and tasks, we remain conservative as the performance differences are quite marginal compared to biomarker prediction differences. This is why we mention that $\text{MV}_{\text{Hybrid}}$ is equal or slightly better than ViTs.

## Appendix E. Eigenvalue Analysis of Pretrained Models

To empirically verify the theoretical analysis, we analyzed the eigenvalues of the four state space/hybrid models. Figure 2 shows the eigenvalue distributions for all four models.

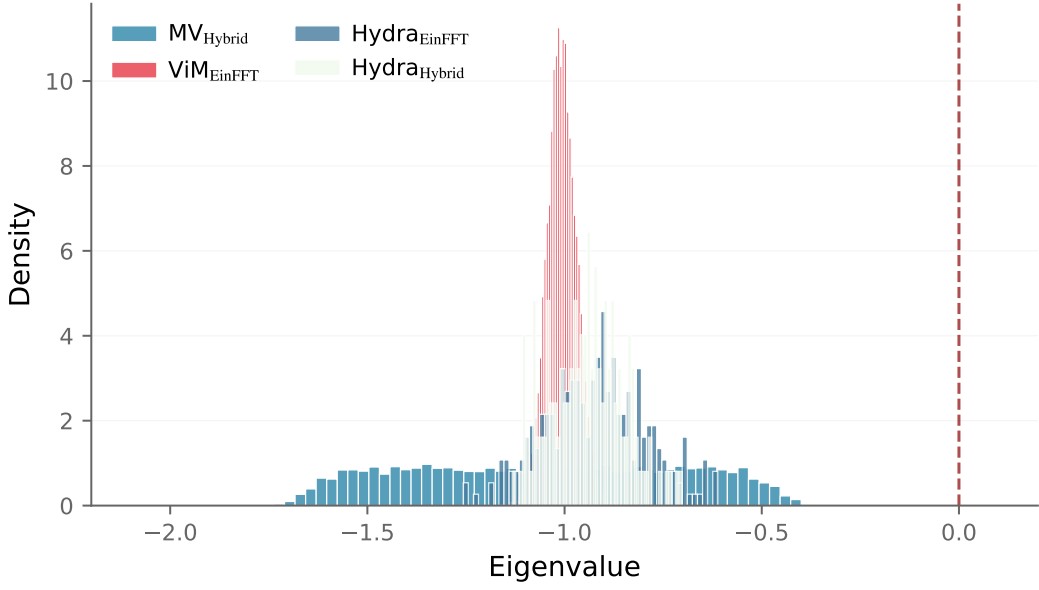

Figure 2: Eigenvalue distributions for pretrained state space models. All models maintain strictly negative eigenvalues through the $A = -\exp(A_{\log})$ parameterization, confirming the enhanced low-frequency bias ideal for biomarker prediction tasks.

As shown in Figure 2, all four state space models maintain strictly negative real eigenvalues through the $A = -\exp(A_{\log})$ parameterization, confirming the theoretical analysis in Section 1.1. MV$_{\text{Hybrid}}$ exhibits the broadest eigenvalue distribution among all models, spanning a wider range of negative values. This broader distribution creates cascaded low-pass filters with diverse cutoff frequencies at $\omega_c = |\lambda_j|$, enabling progressively stronger attenuation of high-frequency components while preserving a richer spectrum of low-frequency features. This broader distribution is most likely due to different initialization schemes, as shown in Appendix A.4. This eigenvalue profile correlates with MV$_{\text{Hybrid}}$'s superior biomarker prediction performance, as the enhanced low-frequency bias captures subtle morphological patterns associated with molecular phenotypes that are critical for accurate gene expression prediction.

