# OpenReview forum: "$MV_{Hybrid}$: Improving Spatial Transcriptomics Prediction with Hybrid State Space-Vision Transformer Backbone in Pathology Vision Foundation Models"
_MICCAI.org/2025/Workshop/COMPAYL — COMPAYL 2025_

### Official Review · Reviewer_6Qqa · 2025-07-13
**MVHybrid: Improving Spatial Transcriptomics Prediction with Hybrid State Space-Vision Transformer Backbone in Pathology Vision Foundation Models**

**Rating:** 5
**Confidence:** 4

**Review:**

## Summary
The authors propose **MV Hybrid**, a hybrid model combining Vision Transformers (ViTs) and state-space models (SSMs) for biomarker prediction tasks in spatial transcriptomics, a rapidly growing field. The study evaluates the model's performance on biomarker prediction, classification, patch retrieval, and survival prediction, demonstrating superior results compared to baseline models like ViT-12, ViT-24, and other SSM-based architectures. The work emphasizes the model's enhanced low-frequency bias, validated through eigenvalue analysis, which improves biomarker prediction by capturing subtle morphological patterns.

- **Clarity**: The paper is well-structured but buries key results (e.g., classification, survival performance) in the appendices, reducing readability.
- **Quality**: The study is rigorous with multiple datasets and tasks, but the evaluation could better highlight computational efficiency or scalability.
- **Originality**: The combination of ViTs and SSMs is not entirely novel, but the initialization schemes and eigenvalue analysis provide empirical validation for low-frequency bias.
- **Significance**: The findings are significant for biomarker prediction in spatial transcriptomics, a hot topic at the moment, with potential broader applications in low-frequency feature extraction.

## Strengths
- **Strong Empirical Validation**: The model is evaluated across multiple tasks, demonstrating consistent performance improvements over baselines.
- **Theoretical Justification**: The eigenvalue analysis provides empirical validation for the model's low-frequency bias, a novel contribution.
- **Outperformance in Biomarker Prediction**: The model shows pronounced improvements in biomarker prediction.
- **Well-Documented Methodology**: The paper includes detailed appendices supporting reproducibility.
- **Open Source**: The authors promise to publish the source code upon acceptance.

## Weaknesses
- **Key Results in Appendices**: Important findings (e.g., eigenvalue distributions, detailed tables) are placed in the appendices, reducing the main text's clarity.
- **Limited Discussion on Efficiency**: The paper does not sufficiently address computational efficiency or scalability compared to baselines.

---

### Official Review · Reviewer_h17D · 2025-07-15
**Transcriptomics Prediction with Hybrid State Space-Vision Transformer**

**Rating:** 3
**Confidence:** 3

**Review:**

The manuscript propose quite complex approach for predicting spatial gene expression from H&E. Authors propose to use  MVHybrid, a hybrid backbone architecture combining state space models (SSMs) with ViT.

Pros:
The study suggest that the MVHybrid offer better linear separability of the embedding.

Cons:
- It is not clear how beneficial this method is for actual predicting spatial transcriptomics from H&E. It would be also great if author show how does this approach compare against standard methods such as multiple-instance learning model for omics prediction from H&E. At the moment the benefits of the study are not very clear.
- It is also not clear how many classes does the model actually predict.

In summary better justification of the proposed method and implication for its use in omics prediction from H&E is needed, as well as comparison with standard methods such as MIL.